# IACRA: Lifetime Optimization by Invulnerability-Aware Clustering Routing Algorithm Using Game-Theoretic Approach for Wsns

**DOI:** 10.3390/s22207936

**Published:** 2022-10-18

**Authors:** Jun Wang, Yadan Zhang, Chunyan Hu, Pengjun Mao, Bo Liu

**Affiliations:** 1School of Information Engineering, Henan University of Science and Technology, Luoyang 471000, China; 2School of Information and Electrical Engineering, China Agriculture University, Beijing 100083, China; 3School of International Education, University of Science and Technology, Luoyang 471000, China

**Keywords:** wireless sensor networks, clustering routing protocol, lifetime, game theory, invulnerability, average residual energy

## Abstract

Energy limitation is one of the intrinsic shortcomings of wireless sensor networks (WSNs), although it has been widely applied in disaster response, battlefield surveillance, wildfire monitoring, radioactivity detection, etc. Due to the large amount of energy consumed for data transmission, how to prolong the network lifespan by designing various hierarchical routing protocols has attracted more and more attention. As a result, numerous achievements have emerged successively. However, these presented mechanisms can rarely guarantee the satisfactory quality of service (QoS), while lowering the energy cost level of WSNs. Meanwhile, invulnerability is undoubtedly an excellent quantitative index to assess QoS. Therefore, it is critical to develop a practical routing method to optimize network lifetime by considering both invulnerability and energy efficiency. Game theory is suitable for such a critical problem as it can be used in node or at network level to encourage the decision-making capabilities of WSNs. In this paper, a novel invulnerability-aware clustering routing algorithm (IACRA) using game-theoretic method is proposed to solve the predicament. The core features of the addressed game-theory-based routing protocol include integral invulnerability awareness, optimal cluster head selection in hierarchical routing, distance-aware cluster head discovery, and cluster rotation update mechanism for lifetime optimization. Particularly, the integral network invulnerability based on weighted fusion is constructed for further defining the profit model by combining the invulnerability indicators used to evaluate the local and whole network. Meanwhile, the optimal probability function of every node elected as CH in per cluster is established through the game between invulnerability and node energy consumption. In addition, the cluster update mechanism base on cluster rotation is proposed to avoid the rapid death of nodes with large energy consumption for maximizing network lifetime. The experimental results indicated a significant improvement in energy balance as well as in invulnerability compared with the other three kinds of well-known clustering routing protocols including GEEC (Game-theory-based energy efficient clustering routing protocol), HGTD (Hybrid, game-theory-based distributed clustering protocol), and EEGC (Efficient energy-aware and game-theory-based clustering protocol). Concretely, at the 400 communication rounds, the invulnerability of IACRA was higher than that of GEEC, HGTD, and EEGC by 77.56%, 29.45% and 15.90%, respectively, and the average residual energy of IACRA was 8.61%, 18.35% and 6.36% larger than that of GEEC, HGTD, and EEGC, respectively. Based on these results, the proposed protocol can be utilized to increase the capability of WSNs against deterioration of QoS and energy constraints.

## 1. Introduction

Wireless sensor networks (WSNs) are an emerging field with a high degree of interdisciplinary and knowledge integration, realizing the functions of sensing monitoring, data processing, and communication transmission in various application scenarios [1,2]. Due to the harsh deployment environment, the node energy cannot be replenished in time and effectively, leading to the energy hole problem in the data collection and transmission procedure and significantly reducing the network lifetime [3]. The network routing protocol can balance the difference in network energy consumption, improve the survival rate of nodes, and boost the network lifetime by optimizing the network topology and data transmission links without increasing hardware cost [4]. Compared with other protocols, the hierarchical routing protocol exhibits considerable advantages in improving energy utilization, reducing transmission delay and packet loss rate, and alleviating the energy hole issue [5]. Inspired by the high reliability and centralized management of hierarchical network architecture, a large number of researchers have extensively investigated cluster head (CH) selection for hierarchical routing protocol. For instance, LEACH (Low energy adaptive clustering hierarchy) protocol generates CHs uniformly through the node bootstrapping mechanism [6]. Zheng et al. proposed an energy-balanced clustering algorithm based on distance to the base station (BS) and neighbor distribution to determine the optimal CHs according to residual energy and neighborhood distribution of nodes [7]. Moreover, Li et al. suggested an energy-balanced routing protocol to divide the network into several clusters by using the *K*-means++ algorithm and select CH by using the fuzzy logical system [8]. However, the current methods of CH selection usually consider the residual energy and location of nodes and ignore the impact of the selection on network invulnerability, leading to inferior invulnerability performance in the face of network attacks. Therefore, combining CH election with invulnerability improvement and energy consumption optimization, selecting CHs with balanced invulnerability and average residual energy is undoubtedly beneficial to extending the network lifetime.

At present, some studies have further changed from the traditional invulnerability quantification analysis to focus on invulnerability enhancement [9]. For example, Fu et al. presented a sink-oriented cascading model for WSNs to resist the cascading failures by topology optimization, and this model can construct a topology with high robustness in a short time [10]. Furthermore, Zhang et al. recommended the fireworks and particle swarm optimization algorithm to optimize the topology structure and elevate the network’s dynamic and static invulnerability [11]. Nevertheless, the invulnerability evaluation indicators used in these existing studies are insufficient and cannot effectively quantify the network invulnerability to provide powerful support for promoting network quality of service (QoS).

In addition, due to the complexity of topology, modeling, and link quality, the success rate of mathematical modelling and competitive analysis for WSNs is unsatisfactory, and game theory has already demonstrated remarkable performance in the optimization techniques for WSNs [12]. Game theory is widely used in the design of routing protocols, including optimal CH selection for hierarchical routing, cluster formation, network topology control, etc. [13]. Pitchai et al. proposed a game-theory-based energy-efficient routing algorithm for the CH election in hierarchical routing to decrease network energy consumption [14]. Moreover, the topology control game algorithm of Markov lifetime prediction model developed by Hao et al. has acceptable capability in ensuring the connectivity and robustness of network [15]. Nonetheless, the existing routing protocols for WSNs based on game theory barely regard the capability of prolonging the network lifetime cycle as a performance criterion and rarely take into account the changes in network invulnerability. Compared with network coverage and connectivity indicators, invulnerability can comprehensively reflect QoS. Hence, the game between invulnerability and average residual energy in the CH election process can balance the energy consumption and improve the ability against external attacks.

To solve the predicaments mentioned above, the objective of this study is to evaluate the effectiveness of an invulnerability-aware clustering routing algorithm (IACRA) using game-theoretic approach for lifetime optimization of WSNs. Firstly, the integration metric describing the invulnerability of the local network and the whole network is proposed to define the profit model, and the energy cost model of nodes becoming CHs or cluster members is established. Secondly, the utility function is set up according to the strategy set to gain the optimal probability of each node elected as CH and then the dominance function of candidate CHs is developed to ensure that the CH with high residual energy and short distance from the BS is obtained. Finally, the cluster update mechanism is employed to rotate the nodes consuming excessive energy to new clusters to balance the power consumption and prolong the network lifetime. In addition, the feasibility and practicality of the proposed method compared to traditional routing protocols are also verified.

The main contributions and novelties of this study include four aspects:By merging the invulnerability indicators used to quantify the local and whole network, the network invulnerability based on weighted fusion is constructed for further defining the profit model;The optimal probability function of each node selected as CH is built through the game between invulnerability and node energy consumption;Based on the residual energy and distance from the BS, the dominance function of candidate CHs is established to ensure the uniqueness of CH per cluster;The performance of invulnerability and average residual energy of WSNs based on the addressed protocol under random, maximum-degree, and maximum-betweenness attacks are systematically analyzed.

The remainder of this paper is arranged as follows. The literature review is presented in Section 2. The network model is given in Section 3. The details of the presented clustering routing algorithm are displayed in Section 4 and Section 5. The validations on the performances of the proposed routing protocol for network lifetime optimization are exhibited in Section 6. Finally, conclusions are shown in Section 7.

## 2. Related Works

WSNs are usually deployed in unattended environments and faced with varied potential attacks, resulting in the increased probability of node failure [16,17]. Thus, the invulnerability improvement has become a vital issue for the practicality and feasibility of WSNs. Due to the restriction of cost and technique, it is impossible to enhance network invulnerability by entirely depending on optimizing node capability. At the same time, improving invulnerability through node backup will seriously boost network redundancy and energy consumption [18]. Therefore, promoting the stability and reliability of WSNs by using efficient network routing schemes without raising the fixed investment has become a trending topic.

The measurement of network invulnerability is undoubtedly a quantitative index to evaluate QoS and is the key to network invulnerability inquisition [19]. Recently, the most widely used invulnerability indicators of non-topological WSNs primarily include network lifetime, survival ratio, and network coverage, etc. [20,21,22]. These indicators are easy to obtain, so one or more indicators are usually used to estimate invulnerability. Since these indexes mainly employ network attributes unrelated to topology structure as the evaluation criteria, these network attributes cannot actually and accurately depict invulnerability [23]. On this basis, the topology-based invulnerability indexes come into being. These indexes apply network connectivity as the object, adjacency matrix as the source of feature extraction, graph theory and probability statistics as theoretical means to compute invulnerability more comprehensively [24,25]. For instance, Chvatalz introduced the graph toughness to assess the invulnerability performance of WSNs [26]. Although there are numerous quantitative studies on topology-based invulnerability, the methods focus on various description perspectives and have specific limitations for different types of networks under the assumption of stationary network topology. However, in the application of hierarchical routing protocols, the clustering structure varies dynamically, resulting in the failure to estimate the invulnerability of whole network in variable environments [27]. The evaluation method based on the equivalent shortest path number can describe the dynamic communication process through the shortest path after the network impairment and can be used to characterize the local invulnerability of network [28]. Moreover, on the basis of connectivity, considering the characteristics of network convergence, the invulnerability measure method using effective connectivity is biased towards the invulnerability of overall network [29]. Thus, it is definitely valuable to combine the invulnerability indicator based on the shortest path number and the invulnerability metric based on the connectivity to clarify the general survivability of network.

Scale-free topology has been used by multiple researchers to improve the invulnerability of WSNs under external attacks [30,31,32,33]. However, these studies fail to precisely assess the impact of different node failure scenarios and damage types on invulnerability performance. In addition, these studies mainly concentrate on appraising network invulnerability, neglecting the analysis and optimization of network structure.

Moreover, the analysis of invulnerability is undoubtedly concerned with the guarantee of QoS performance under the impact of various network attacks [34,35,36,37]. Nevertheless, the related studies cannot fully reflect the difference value of network invulnerability in the process of node failure. Furthermore, it can effectively indicate the dynamic variation of invulnerability by describing the change in network performance caused by removing critical nodes [38]. However, the existing attack schemes rarely involve the evaluation of node importance, which is usually called centrality. Jahanpour discovered that compared with other centrality indicators (reciprocal closeness, complement-derived closeness, eigenvector centrality, etc.), degree and betweenness could be used more satisfactorily to select important nodes [39]. Therefore, in this study, two heuristic node deliberate attack strategies are designed using degree and betweenness. In addition, the failure of nodes under network attack belongs to a transition process in practice, and the nodes do not die immediately. Accordingly, the arbitrary removal of these nodes does not conform to the actual circumstances. In this paper, we presume that the energy of nodes will decline significantly once attacked.

For balancing invulnerability and energy consumption in WSNs, game theory has become a unique method for developing routing protocols, and its rational behavior can be applied to the systematic analysis of hierarchical networks [40]. The routing protocols based on game theory have demonstrated more attractive advantages than other protocols regarding network lifetime, energy efficiency, and route establishment time [41]. Considering the factors such as residual energy and link quality, the loss function and profit function can supply proper decisions in CH competition, simplifying the optimal CH selection process for cluster-based routing protocols [42]. Lin et al. utilized an innovative routing protocol named game-theory-based energy efficient clustering routing protocol (GEEC) to reduce average energy consumption [43]. Wu et al. employ an efficient energy-aware and game-theory-based clustering protocol (EEGC) to select CHs through equilibrium probability of the game and residual energy dependence index [44]. Yang et al. used a hybrid, game-theory-based distributed clustering protocol (HGTD) for getting an equilibrium probability by playing a localized clustering game for each node [45]. The advantages and disadvantages of the above-mentioned three routing protocols are summarized in Table 1. However, these studies on routing protocols based on game theory rarely integrate QoS indexes into the game process and generally explore network performance through mixed routing indicators. Therefore, QoS cannot be ensured under external attacks because network invulnerability is not involved in the definition of the game model.

## 3. Preliminaries

The network model used in this study is a WSN model in which N nodes are randomly allocated in a square sensing area (*L* × *L*) centered on a BS. The BS has strong computing and network management capabilities and is equipped with more battery power or can be self-replenished through energy harvesting. Therefore, the BS has no energy limitation. On this basis, the following assumptions are made about the WSN.
Each node has a unique identity document (ID), and all the nodes are evenly distributed in the monitoring area;Nodes maintain stationary after the network deployment, and their energy is restricted and cannot be replenished;The BS cannot move and can communicate with any node in the detection area;Each node has multiple adjustable transmitting capacities and can choose the appropriate transmission power in terms of the communication distance;The network adopts data fusion technology to decrease the amount of data transmission;Every node does not have position perception capability;Under a certain transmission power, an individual node can calculate the communication distance according to the received signal strength;The simplified model proposed by Heinzelman et al. [46] is adopted in our study for computing communication energy consumption in consideration of path loss.

## 4. Game Clustering Model

### 4.1. Definition of Game Space

Participant: The active nodes in WSNs, denoted as *N* = {1, 2, 3, …, *N*}.

Strategic space: The actions of nodes are carried out in rounds. In each round, nodes select respective strategies from the strategy set *S =* {*D*, *ND*}, *D* denotes that the node participates in the CH election, and *ND* implies that the node does not partake in the CH election.

Utility function: The strategies of participants are interdependent, and the benefits of each participant are related to the strategy set of other participants. Accordingly, *U =* {*U*(*i*), *i* = 1, 2, 3, …, *N*} is used to depict the node profit set corresponding to the strategy set in this study.

In the network, CHs process the data collected from ordinary nodes and transmit it to the BS in the form of data packets. Meanwhile, nodes can choose strategy *D* or strategy *ND* to maximize their payoffs. If no node chooses strategy *D* to become a CH and the data acquired by nodes cannot be sent to the BS, the profits of all nodes equal 0. In the case that at least one node *j* selects strategy *D* to be a CH, then the payoffs of node *j* and any node other than node *j* are $V\left(i\right)-C_{CH}\left(i\right)$ and $V\left(i\right)-C_{CM}\left(i\right)$, respectively, where V(i) is the gain of the data collected by the node successfully reaching the BS, $C_{CH}\left(i\right)\;$ is the loss of the node elected as CH, $C_{CM}\left(i\right)$ is the punishment for the node becoming an ordinary node. In the case that $C_{CM}\left(i\right)>C_{CH}\left(i\right)$, $V\left(i\right)-C_{CH}\left(i\right)>V\left(i\right)-C_{CM}\left(i\right)$, being elected as CH can obtain more benefits, and all nodes will definitely choose the CH strategy. Therefore, in this paper, the following assumption $C_{CM}\left(i\right)<C_{CH}\left(i\right)<V\left(i\right)$ is set. Furthermore, the profit function *U*(*i*) of any node *i* is defined as
(1)$$U\left(i\right)=\left\{\begin{array}{ll} 0 & if\;s_{j}=ND,\forall j\in N\\ V\left(i\right)-C_{CM}\left(i\right) & if\;s_{j}=D\;\\ \left(i\right)-C_{CH}\left(i\right) & if\;s_{j}=ND\;and\\ & \;\;\;\;\exists j\in N,s.t.s_{j}=D \end{array}\right.$$

The corresponding profit matrix is shown in Table 2.

Based on the above matrix, in the situation that all nodes are selected as CHs or all nodes are elected as ordinary nodes, both the strategy set *S* = {*D*, *D*, …, *D*} and *S* = {*ND*, *ND*, …, *ND*} are not a Nash equilibrium. The overall payoff is the most satisfactory when one node becomes CH and other nodes are ordinary nodes.

### 4.2. Profit and Loss Analysis

#### 4.2.1. Profit Model

Invulnerability based on the number of shortest paths

**Definition** **1.***For a network with N nodes, if there are u_0_(k_min_) shortest paths with the number of hops equaled k_min_ between node i and node j, then the number of equivalent shortest paths between node i and node j is expressed as em_ij_, which can be computed by Equation (2)*.
(2)$$\left\{ \begin{array}{l} u_{0}\left(k_{min}\right)=\frac{\left(N-2\right)!}{\left(N-k_{min}-1\right)!}\\ u_{1}\left(k_{min}\right)=\overset{k_{min}}{\underset{t=1}{\sum }}\frac{\left(N-2\right)!}{\left(N-t-1\right)!}\\ \;\;\;\;\;\;\;em_{ij}=\frac{u_{0}\left(k_{min}\right)}{u_{1}\left(k_{min}\right)} \end{array}\right.$$
where the number of hops *k_min_* ≤ *N*−1, *u*_1_(*k_min_*) refers to the number of paths in which the number of hops between nodes is less than *k_min_* in the corresponding *N*-node fully-connected network, and *t* indicates an integer variable. Obviously, *em_ij_* ∈ (0,1], in the condition that there is a direct edge between two nodes, *em_ij_* is equal to 1. For any node *i* and node *j*, if *em_ij_* is equivalent to 1, the network is regarded as fully connected.

The invulnerability of each cluster in the network is equal to the equivalent shortest path number of the cluster and is denoted as follows
(3)$$inv=\frac{\sum_{i=1}^{N-1}\sum_{j=i+1}^{N}em_{ij}}{N\left(N-1\right)/2}$$
where *N* signifies the number of nodes in a cluster, $\overset{N-1}{\underset{i=1}{\sum }}\overset{N}{\underset{j=i+1}{\sum }}em_{ij}$ is the sum of the number of equivalent shortest paths in the entire cluster, and *N*(*N* − 1)/2 is the number of node-to-node connections that can be established. For an incompletely connected network, the equivalent shortest path number *inv* ∈ (0,1), it can be easily noticed that the larger the *inv*, the more compact the cluster structure and the more potent the invulnerability.

**Definition** **2.**
*For a network composed of *

ω

* clusters, the invulnerability of the network can be measured by the weighted sum of the invulnerability per cluster. The invulnerability indicator of the network INV can be calculated as follows*

(4)
$$INV=\overset{\omega }{\underset{m=1}{\sum }}\frac{N_{m}}{N}inv_{m}$$

*where N_m_ is the number of nodes in the mth cluster, N is the number of nodes in the whole network, and inv_m_ is the invulnerability of the mth cluster.*


2.Invulnerability based on connectivity

The invulnerability measurement based on effectively connected clusters is defined as follows
(5)$$C=\frac{1}{\omega \sum_{m=1}^{\omega }\frac{N_{m}}{N}\delta_{m}l_{m}}$$
where ω  represents the number of the clusters in the network, $N_{m}\;$ means the number of nodes in the *m*th cluster, *N* is the total number of nodes in the network, and *l_m_* is the average length of the shortest path between any nodes in the *m*th cluster.
(6)$$\delta_{m}=\{\begin{array}{l} 1\\ 0 \end{array}\begin{array}{l} \;\mathrm{Nodes}\;\mathrm{of}\;N_{m}\;\mathrm{at}\;\mathrm{least}\;\mathrm{has}\;\mathrm{one}\;\mathrm{link}\;\mathrm{with}\;\mathrm{BS}\\ \;\mathrm{None}\;\mathrm{of}\;\mathrm{the}\;\mathrm{nodes}\;\mathrm{of}\;N_{m}\;\mathrm{has}\;\mathrm{link}\;\mathrm{with}\;\mathrm{BS} \end{array}$$

The invulnerability based on the number of shortest paths focuses on the survivability of the local network. On the other hand, the invulnerability based on connectivity is biased towards the destruction-resistant of the whole network. Thus, by combining the two indicators, the network invulnerability in the case that node *i* is elected as CH can be expressed as follows
(7)$$B_{i}=\lambda \times INV+\left(1-\lambda \right)\times C\;,\;0\leq \lambda \leq 1,\;1\leq i\leq N$$

In the study, the utility function of the game model is described as the quadratic function of the network invulnerability *B*(*i*) and the residual energy of the node *E*(*i*), then the utility function in the game model of CH selection is defined as
(8)$$\{\begin{array}{l} U\left(B\right)=\overset{N}{\underset{i=1}{\sum }}B_{i}e_{i}-\frac{1}{2}\left(\overset{N}{\underset{i=1}{\sum }}B_{i}^{2}+2\rho \overset{N}{\underset{i\neq j}{\sum }}B_{i}B_{j}\right)-\overset{N}{\underset{i=1}{\sum }}E_{i}B_{i}\\ e_{i}=1-\frac{N_{m}}{N} \end{array}$$
where ρ(0≤ρ≤1) expresses the competitive factor for nodes to be selected as CHs. The circumstance that ρ = 1 implies no difference in the invulnerability, whether the node is selected as CH or not. In the case of ρ = 0, it means that the node is assigned as CH, which has an irreplaceable advantage of invulnerability. *B_j_* is the invulnerability when another node except node *i* is selected as CH, 1≤j≤N, *e_i_* is the efficiency factor of *B_i_* corresponding to node *i*, *N_m_* is the number of nodes in the *m*th cluster, and *N* is the total number of nodes in the network.

The optimal invulnerability *B_i_* can be obtained by maximizing *U*(*B*). Accordingly, we take the derivative of *U*(*B*) with respect to *B_i_* and set it to 0, as follows
(9)$$\frac{\partial U\left(B\right)}{\partial B_{i}}=e_{i}-B_{i}-\rho \overset{N}{\underset{i\neq j}{\sum }}B_{j}-E_{i}=0$$

Consequently, the invulnerability function can be derived as follows
(10)$$B_{i}\left(E\right)=\frac{\left(e_{i}-E_{i}\right)\left[\rho \left(N-2\right)+1\right]-\rho \sum_{i\neq j}\left(e_{j}-E_{j}\right)}{\left(1-\rho \right)\left[\rho \left(N-1\right)+1\right]}$$

In addition, *V_i_* is the profit function, which can be described as
(11)$$V_{i}=B_{i}\left(E\right)\times \rho +\frac{E_{i}}{E}\left(1-\gamma \right)$$
where *B_i_*(*E*) is the network invulnerability if node *i* is selected as CH, *E_i_* is the current remaining energy of node *i*, *E* is the initial energy of node. Meanwhile, *γ* is an adjustable parameter. If the node is not attacked, *γ* equals 1; otherwise, *γ* ∈ [0,1).

#### 4.2.2. Cost Model

The loss of selected CH *C_CH_*(*i*) primarily includes receiving data packets from cluster members, fusing data packets of cluster members, and the energy consumed by transferring data packets to the BS. In this study, the free space channel is assumed to be utilized for intra-cluster communication, and the multipath fading channel is presumed to be used for communication between CHs and the BS. Thus, *C_CH_*(*i*) is defined as
(12)$$C_{CH}\left(i\right)=C_{Tx\left(CH,BS\right)}+C_{Rx\left(CH,i\right)}+C_{aggr}$$
where *C_Tx_*_(*CH*,*BS*)_ is the energy required to transmit data packets from CH to the BS, *C_Rx_*_(*CH*, *i*)_ is the energy consumed by receiving data of cluster members, *C_aggr_* is the energy consumption of data fusion through CH, and *C_Tx_*_(*CH*,*BS*)_ is given by
(13)$$C_{Tx\left(CH,BS\right)}=kd_{\left(CH,BS\right)}^{2}\varepsilon_{mp}+kE_{elec}$$
where *d*_(*CH*,*BS*)_ represents the distance between CH and the BS, and *ε_mp_* denotes the power amplification constant under the multipath fading channel model. *E_elec_* is the power consumption of sending and receiving 1 bit data, and *k* is the packet size. Moreover, the energy consumption of the packets acquired and fused by CH is proportional to the packet size. Hence, *C_Tx_*_(*CH*,*i*)_ and *C_aggr_* are obtained by
(14)$$C_{Tx\left(CH,i\right)}=\left(N_{m}-1\right)kE_{elec}$$
(15)$$C_{aggr}=kE_{aggr}$$
where *N_m_* is the number of nodes in the cluster, and *E_aggr_* is the energy consumption of merging data of a node.

Therefore, the cost of the selected CH is described by
(16)$$C_{CH\left(i\right)}=kd_{\left(CH,BS\right)}^{2}\varepsilon_{mp}+N_{m}kE_{elec}+kE_{aggr}$$

In addition, the cost of cluster members is depicted by
(17)$$C_{CM}\left(i\right)=kd_{\left(CH,i\right)}^{4}\varepsilon_{fs}+kE_{elec}$$
where *d*_(*CH*,*i*)_ is the distance between cluster member *i* and CH, and *ε_fs_* is the power amplification constant of free space channel model.

### 4.3. Utility Function

The game model includes two strategies, *D* and *ND*, and the probabilities of choosing strategy *D* and *ND* for a node are assigned as *P* and 1 − *P*, respectively. We assume that the cluster that node *i* locates at has *N_m_* nodes, each node competes with other *N_m_*_−1_ nodes, and the cluster payoff relies on the number of choices of strategy *D*. If strategy *ND* is selected for all *N* nodes, the benefit for all nodes is 0. Suppose at least one node chooses strategy *D* and the income of the node is *V*(*i*) − *C_CH_*(*i*), then the number of nodes that select strategy *ND* is *N_m_*_−1_ and the revenue of each node is *V*(*i*) − *C_CM_*(*i*). The total payoff of these *N_m_*_−1_ nodes can be expressed as follows:(18)$$U_{ND}\left(i\right)=\left[V\left(i\right)-C_{CM}\left(i\right)\right]\left[1-{\left(1-p\right)}^{N_{m}-1}\right]$$

Furthermore, the average income of node *i* is obtained by
(19)$$\begin{array}{ll} \bar{U\left(i\right)} & =p\times U_{D}\left(i\right)+\left(1-p\right)\times U_{ND}\left(i\right)\\ & =p\left[V\left(i\right)-C_{CH}\left(i\right)\right]+\left(1-p\right)\left[V\left(i\right)-C_{CM}\left(i\right)\right]\left[1-{\left(1-p\right)}^{N_{m}-1}\right] \end{array}$$

Subsequently, ${\bar{U\left(i\right)}}^{\prime }$ is achieved by taking the derivative of $\bar{U\left(i\right)}$ to compute the maximum benefit as follows
(20)$${\bar{U\left(i\right)}}^{\prime }=C_{CM}\left(i\right)-C_{CH}\left(i\right)+N_{m}\left[V\left(i\right)-C_{CM}\left(i\right)\right]{\left(1-p\right)}^{N_{m}-1}=0$$
(21)$$p_{i}^{*}=1-{\left(\frac{C_{CH}\left(i\right)-C_{CM}\left(i\right)}{N_{m}\left[V\left(i\right)-C_{CM}\left(i\right)\right]}\right)}^{\frac{1}{N_{m}-1}}$$
where $p_{i}^{*}$ is the optimal probability function for node *i* to select strategy *D*.

## 5. Routing Scheme and Attack Types

### 5.1. Communication Flow

The presented clustering routing algorithm based on game theory can be separated into three stages: initialization, cluster establishment, and stable communication, as shown in Figure 1. Specifically, the initialization stage mainly gathers the network details in the monitoring area, such as node location, energy, ID, etc.; the cluster establishment stage comprises CH selection and cluster formation; the stable communication stage implements the information transmission from cluster members to CHs and transfers the fused data to the BS.

Initialization phase

First, the BS adopts flooding mode to broadcast the ‘networking’ message to the nodes within the monitoring area. Each node documents the distance from the BS and adjusts the appropriate transmitting power for dispatching individual information. Based on the location information, the BS divides the entire area into *ω* clusters, and *ω* is computed as follows
(22)$$\omega =\sqrt{\frac{N}{2\pi }}\frac{L}{\bar{d}}$$
where $\bar{d}$ is the average distance between all nodes and the BS.

Secondly, according to the node information from each cluster, the BS calculates the invulnerability and cost of each node as CH and the cost of becoming a common node.

Finally, the BS computes the optimal probability *p_i_** that each node in every cluster becomes CH and transfers the probability data of individual cluster to the local information table per node in the cluster for storage;

2.Cluster establishment phase

At this stage, every node in the *ω* clusters has three states: common node, candidate CH, and CH. These nodes with the highest probability are nominated as the candidate CHs for each cluster. Assuming that merely one candidate CH is in a cluster, the node declares itself as CH; Moreover, in the case that there is more than one candidate CH, the dominance function *g_opt_*(*i*) of all candidate CHs in a cluster is calculated by
(23)$$g_{opt}\left(i\right)=\frac{E_{i}}{E}\times \frac{1}{{\left(d^{\prime }\right)}^{2}}$$
where $d^{\prime }\;$ is the distance between CH and the BS.

Then, the candidate CH with the highest dominance function value is designated as CH. If multiple candidate CHs have the same maximum value, the candidate CH with the smallest ID is assigned as CH. Furthermore, after completing the CH selection, each chosen CH sends a clustering message containing the ID of CH to the nodes in the corresponding cluster. The common nodes in a cluster return a reply message to CH to apply for cluster membership following the reception of the message. Until all nodes in the monitoring area join the clusters, the cluster establishment stage terminates;

3.Stable communication phase

During the phase, CHs send the TDMA scheduling table to the corresponding cluster members to coordinate the data transmission for avoiding data conflict. Following the information acquisition from cluster members, CHs implement data fusion to further transmit the data packets to the BS in single-hop mode.

### 5.2. Cluster Update Mechanism

The network re-conducts the CH election when the average energy of the whole network declines by *e*%. Particularly, nodes that have been elected as CHs will quit the new CH election until all nodes in the cluster are selected as CHs. The total number of nodes participating in the game simultaneously should be the total number of nodes minus the number of elected CHs.

Each cluster rotates π/*ω* counterclockwise after *r* CH elections to modify the cluster location and cluster members. In this way, the nodes that consume excessive energy in the original cluster can be included in the new cluster, reducing the likelihood of these nodes becoming CHs and thus prolonging the node lifetime.

### 5.3. Strategies of Network Attacks

In order to comprehensively assess the performance of WSNs against different attacks, the designed attack scheme mainly have random attack and deliberate attack in our study. The random attack means that the probability of nodes attacked in each cluster is without distinction, while the deliberate attack is the selective attack of some nodes in per cluster on the basis of a specific strategy.

Random Attack

Under the random attack strategy, the nodes with a proportion of *p*^1^ are stochastically selected from the initial network, resulting in a drop in node power;

2.Deliberate Attack

Under the maximum-degree attack scheme, the proportion of nodes attacked in descending order of degree centrality is *p*^2^, and the attack causes a decline in node energy. The equation to calculate degree centrality in a cluster with *N_m_* nodes is defined by
(24)$$C_{d}\left(x\right)=\frac{h\left(x\right)}{N_{m}-1}$$
where *N_m_* is the number of nodes in the *m*th cluster, and *h*(*x*) is the number of direct connections between node *x* and other *N_m_* − 1 nodes in the cluster.

For the maximum-betweenness attack scenario, the ratio *p*^3^ of nodes is attacked in descending betweenness centrality, inducing the reduction in node energy. The formula to computer betweenness centrality for a cluster is given by
(25)$$C_{c}\left(x\right)=\frac{\sum_{j\in N_{m}}g_{j}\left(x\right)/g_{j}}{N_{m}-2}$$
where *N_m_* expresses the number of nodes in the *m*th cluster, *g_j_*(*x*) indicates the number of shortest paths from node *j* to CH through node *x*, and *g_j_* implies the number of shortest paths from node *j* to CH.

## 6. Results and Analysis

### 6.1. Initialization of Simulation P arameter

In this section, the nodes are deployed in a square area with a side length of 100 m, the number of nodes *N* ranges from 800 to 1200, the node decline degree *e*% varies from 5% to 25%, and the number of CH elections *r* alters between 1 and 5. In addition, the data packet length *k* is set to 2000, 3000, 4000, 5000 and 6000, respectively. The termination condition of simulations is that the death rate of nodes reaches 80%. The detailed initial simulation parameter settings are shown in Table 3.

After the initialization of the simulation test, we assume that the three categories of attacks assail the network. Proportions *p*^1^, *p*^2^, and *p*^3^ are set to 10%, and the energy of nodes under attack is presumed to be reduced to 0.25 J. Proceed to the next step, the variation regulations of invulnerability and average residual energy caused by the change of network parameters under various attack modes are investigated in detail.

### 6.2. Impact of Network Parameters on Invulnerability

Figure 2 depicts the difference in invulnerability under three attack strategies with varied *e*%. For different *e*%, the invulnerability of the three attacks reaches the maximum when the number of communication rounds is 101. At this point, the descending sorting of invulnerability is maximum-degree attack, maximum-betweenness attack, and random attack. Moreover, under the case of *e*% = 10%, 15%, 20%, and 25%, respectively, and the number of communication rounds equals 400, the order of invulnerability for the three attack modes remains identical. In contrast, at *e*% = 5%, the invulnerability of maximum-betweenness attack (0.350 J) is the largest and that of random attack (0.313 J) is the least.

Figure 3 demonstrates the changes in invulnerability under three attacks with varying *k*. The invulnerability subjected to these three attacks reaches the maximum at the same number of communication rounds (101). In comparing the maximum invulnerability, it is easy to observe that the invulnerability of the maximum-degree attack is the highest, while that of the random attack is the lowest. In particular, the greatest invulnerability (0.995 J) of the maximum-degree attack is the most remarkable for *k* = 6000. In the circumstance that the number of communication rounds is 400, at *k* = 2000, the invulnerability of maximum-degree attack substantially surpasses that of random attack (64.19%); at *k* = 5000, the invulnerability of maximum-degree attack is superior to that of maximum-betweenness attack and the invulnerability of maximum attack is 23.89% larger than that of random attack.

Figure 4 shows the shifts in invulnerability under three attacks with various *r*. When *r* is 1, 3, 4, and 5, the invulnerability of the three attacks simultaneously attains the maximum at 21, 61, 81, and 101 communication rounds, respectively. At *r* = 2, the invulnerability of random attack arrives at the peak value when the number of communication rounds is 21. Furthermore, the invulnerability of maximum-degree and maximum-betweenness attacks reaches the greatest value at 41 rounds. In the condition that *r* is 2, 3, and 5, it is obviously found that the invulnerability of the maximum-degree attack is the most satisfactory. At the same time, when *r* is 1 and 4, it is easily discovered that the maximum-betweenness attack has the most excellent invulnerability, but the random attack has the lowest invulnerability in both cases. In addition, in the case that *r* is 2, 3, 4, and 5, the invulnerability of maximum-degree attack is the most outstanding, and that of random attack is the most inferior; at *r* = 1, the invulnerability of maximum-betweenness attack is more heightened than that of maximum-degree attack and random attack.

Figure 5 exhibits network invulnerability changes with different *N* for the three attacks. At 101 rounds, it can be uncovered that the highest invulnerability of maximum-degree, maximum-betweenness, and random attacks is ranked from high to low. For *N* = 1200, the most prominent invulnerability (0.997 J) of maximum attack is the highest. Moreover, when *N* is 1200 and 1000, the utmost invulnerability of maximum-betweenness attack is 0.990 J and 0.733 J, respectively, and the former is 35.06% higher than the latter. Additionally, in the case of *N* = 1200, the maximum damage resistance of random attack (0.822 J) is 25.69% higher than that of *N* = 800. Especially, the invulnerability order of the three attacks at 400 rounds is the same as that at 101 rounds.

To sum up, we can apparently notice that the order of invulnerability under the three attacks is inconsistent with the more robust capability of scale-free networks to resist random attacks with the variable network parameters. It may be that after critical nodes are attacked, CHs with improved invulnerability and balanced energy consumption can be selected according to the optimal probability of CH selection based on invulnerability and residual energy, providing the optimization strategy for subsequent network clustering and communication. Meanwhile, the cluster update mechanism can suppress the invalid network topology caused by the failure of nodes with higher degree centrality and betweenness centrality. In addition, with the increase of *r*, the number of communication rounds rises. It is because a less *r* implies that the network needs to be re-networking more frequently, leading to a considerable decrease in the remaining energy of the whole network. Compared with random and maximum-betweenness attacks, the proposed routing algorithm can sufficiently decrease the damage to the network provoked by the maximum-degree attack, improve the network invulnerability and positively impact network service performance.

### 6.3. Impact of Network Parameters on Average Residual Energy

Figure 6 exhibits the change of the average residual energy under the three attacks with altering *e*. In the case of various *e*, the maximum-degree and maximum-betweenness attacks decline rapidly in the later communication period. In addition, when *e* is 5, 10, and 20, respectively, the average residual energy of random attack is larger than that of maximum-degree attack for the whole communication process. In the condition that *e* is 5, 10, 15, 20 and 25, respectively, we can distinctly perceive that the average residual energy of random attack is bigger than that of maximum betweenness attack within the interval of [1,516], [1,512], [1,536], [1,665], and [1,494], respectively. If *e* is 5, 10, 15, and 20, respectively, the average residual energy of the maximum-degree attack is greater than the maximum-betweenness attack in the early stage but less than the maximum-betweenness attack in the later phase. However, in the case of *e* = 25, the average residual energy of the maximum-betweenness attack is consistently higher than that of the maximum-degree attack.

Figure 7 shows the variation characteristics of average residual energy under these three attacks with varied *k*. Significantly, both the maximum degree and the maximum betweenness attacks lower sharply in the later phase for diverse *k*. When *k* is set to 4000, 5000, and 6000, the average residual energy of random attacks is greater than the maximum-degree attack. Furthermore, if *k* equals 5000 and 6000, the average residual energy of maximum-betweenness attack will be higher than that of maximum-degree attack.

Figure 8 displays the variation of average residual energy under the three attacks with differing *r*. It can be evidently seen that the maximum-degree attack and the maximum-betweenness attack decline enormously with the growth in communication rounds. Meanwhile, in the situation that *r* is 1, 2, 3, and 5, respectively, it is discovered that when the number of communication rounds is in a specific interval, the average residual energy of random attack is smaller than that of maximum-betweenness attack. Moreover, other features worthy of our attention are as follows: when *r* is 3, 4, and 5, the average residual energy of random attack is bigger than the maximum-degree of attack; in the case of *r* = 4, the average residual energy of random attack is greater than the maximum-betweenness attack; when *r* is 1, 2, and 3, the average residual energy of maximum-betweenness attack is larger than the maximum-degree attack.

Figure 9 illustrates the changes in average residual energy for the three attacks with different *N*. The stability of average residual energy of maximum-degree and maximum-betweenness attacks is obviously worse than that of random attack with the increment of communication rounds. It should be noted that in the circumstance that *N* is 800 and 1000, the average residual energy of random attack is greater than that of maximum-degree attack. When *N* is 800, 1100, and 1200, the average residual energy of maximum-betweenness attack is more excellent than that of maximum-degree attack.

In summary, under the condition of assorted network parameters, we can observe that the average residual energy of random attack reduces the slowest and that of maximum-degree attack declines the promptest. This result is consistent with the opinion that the power consumption provoked by maximum-degree and maximum-betweenness attacks is more remarkable than that of random attack. Therefore, in the case of the random attack, the advantages of the routing algorithm are higher than that of the maximum-degree and maximum-betweenness attacks, providing an effective solution to network energy balance. Moreover, high-degree centrality nodes have more adjacent nodes, and more shortest paths pass through high-betweenness centrality nodes than ordinary nodes. These nodes play a more crucial role in the network, and the attack targets of maximum-degree attack and maximum-betweenness attack are these necessary nodes. However, the failure of essential nodes will induce frequent network route reconstruction, and the average residual energy of the network decreases faster. In addition, the rapid decline of the average residual energy may generate energy holes, so the average residual energy of maximum-degree attack and maximum-betweenness attack drops promptly at a larger number of communication rounds.

### 6.4. Algorithm Comparison

Figure 10 and Figure 11 illustrate the comparison results in terms of invulnerability and average residual energy between the proposed IACRA and GEEC, HGTD, and EEGC algorithms. It can be distinctly seen that the number of communication rounds of the four algorithms is 816, 727, 676, and 762, respectively. As shown in Figure 10, IACRA, GEEC, HGTD, and EEGC obtain the maximal invulnerability at the same number of communication rounds, which is 0.992, 0.638, 0.990, and 0.918, respectively. At 400 rounds, the invulnerability of IACRA is higher than that of GEEC, HGTD, and EEGC by 77.56%, 29.45%, and 15.90%, respectively. Concretely, the invulnerability of HGTD is greater than that of IACRA only when the number of communication rounds is within [1,101]. With the growth in the number of communication rounds, the invulnerability of HGTD drops faster and is gradually lower than that of IACRA, GEEC and EEGC. During the entire communication process, the invulnerability of IACRA is superior to that of GEEC and EEGC.

Furthermore, as displayed in Figure 11, the average residual energy of IACRA is 8.61%, 18.35%, and 6.36% larger than that of GEEC, HGTD, and EEGC at 400 rounds, respectively. IACRA, GEEC, HGTD, and EEGC rapidly decline when communication rounds are 753, 663, 622, and 729, respectively. At this moment, the average residual energy corresponding to these four algorithms is 0.131 J, 0.133 J, 0.127 J, and 0.119 J. The above performance is caused by the fact that GEEC, HGTD, and EEGC merely pay attention to the impact of energy consumption, distance between nodes and the BS, and degree of nodes on game equilibrium in calculating CH selection probability. However, the CH election procedure of IACRA takes into consideration the balance of invulnerability and energy-consumption, and the distance between nodes and the BS is considered in the definition of the dominance function for CH selection, which is beneficial to making the candidate CH with high residual energy and that closer to the BS to become CH. Meanwhile, the presented cluster updating mechanism can rotate the nodes with low residual energy into new clusters, thus dramatically lengthening the network lifetime. In this study, compared with GEEC, HGTD, and EEGC, a novel invulnerability indicator is introduced to describe the network’s QoS sufficiently. Meanwhile, the game-theoretic selection mechanism of CHs balances the invulnerability and residual energy and enhances the robustness of network topology. In addition, the cluster update mechanism based on cluster rotation extends the lifetime of essential nodes and reduces the network energy consumption.

## 7. Conclusions

In this paper, the invulnerability-aware clustering routing algorithm based on game-theoretic method is proposed to improve invulnerability and balance network energy consumption in WSNs. In the network, each node in an individual cluster achieves the optimal probability of becoming a CH via the game of invulnerability and residual energy. In addition, the dominance function for candidate CHs considering the remaining energy and the distance from the BS is constructed to avoid the scenario of multiple CHs in a cluster. A cluster update mechanism is established to further optimize the network energy consumption by rotating the nodes with excessive energy consumption to new clusters. Furthermore, the network invulnerability under three external attacks rises first and then declines with varied network parameters. Specifically, the invulnerability of maximum-degree attack is the highest, indicating that the addressed routing algorithm has the most robust capability to reduce the negative impact on invulnerability performance of maximum-degree attack. With the change of network parameters, the average residual energy of random attack decreases the slowest, signifying that the presented routing algorithm can achieve the most satisfactory energy performance when encountering random attack. At the same time, increasing the number of CH elections *r* is useful to significantly boost the number of communication rounds.

For the proposed routing protocol, CH election computation and intra-cluster communication overhead mainly concentrated in the clustering setup and maintenance phase. Therefore, in future research, to reduce energy consumption, more representative nodes should be preliminarily chosen to participate in the CH election according to the distance from the BS, residual energy, node centrality, etc. Meanwhile, the weight of invulnerability and residual energy in the profit function should be adaptively adjusted to obtain more acceptable network performance under network attacks. In addition, IACRA is superior to GEEC, HGTD, and EEGC in terms of extending the network life cycle, improving the network invulnerability and balancing the network energy consumption. Nevertheless, the feasibility and effectiveness of the routing protocol in other network topologies still require verification. Furthermore, in the case of introducing new adversaries, rebuilding the profit matrix, modifying the existing game model, and reconducting a Nash equilibrium are undoubtedly demanded.

## Figures and Tables

**Figure 1 sensors-22-07936-f001:**
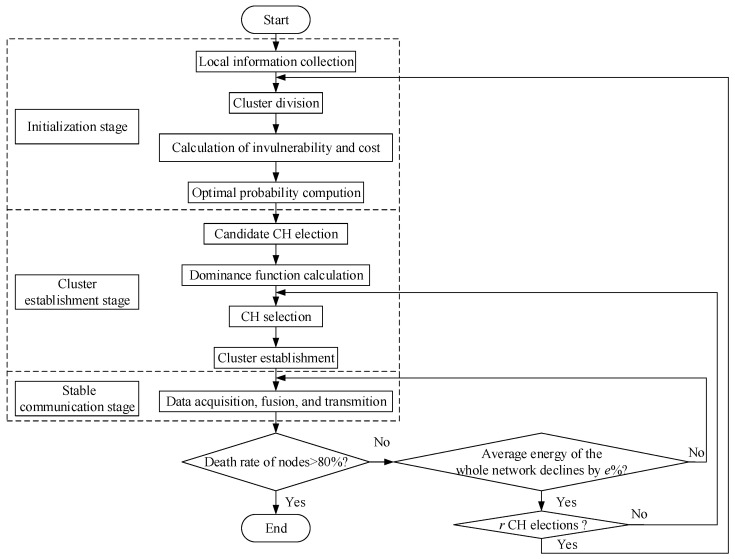
Flowchart of routing scheme.

**Figure 2 sensors-22-07936-f002:**
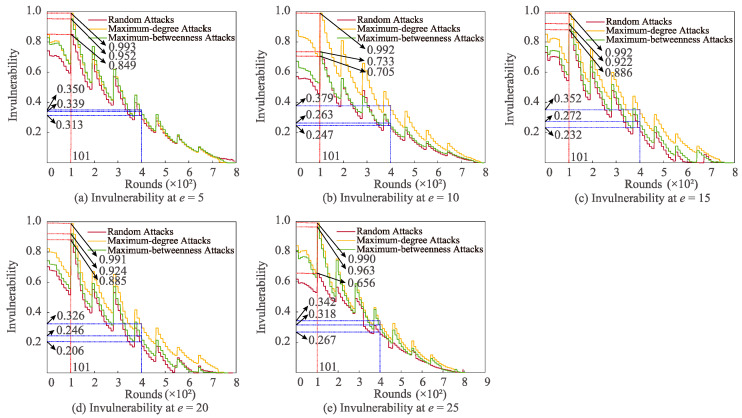
Relationships between differing *e* and invulnerability under distinct attacks.

**Figure 3 sensors-22-07936-f003:**
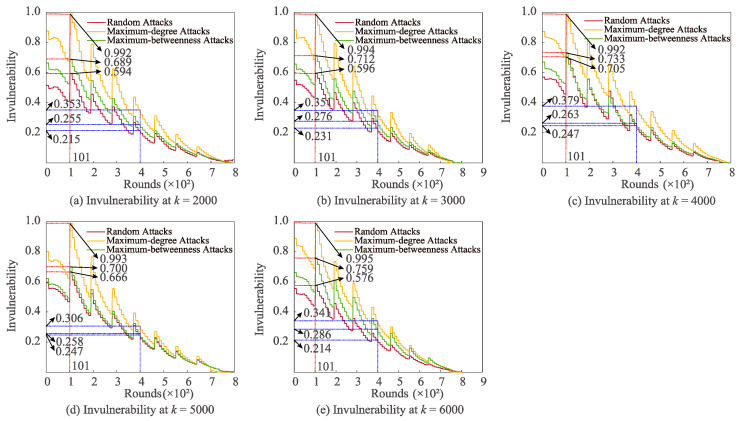
Effects of changing *k* on invulnerability for varied attacks.

**Figure 4 sensors-22-07936-f004:**
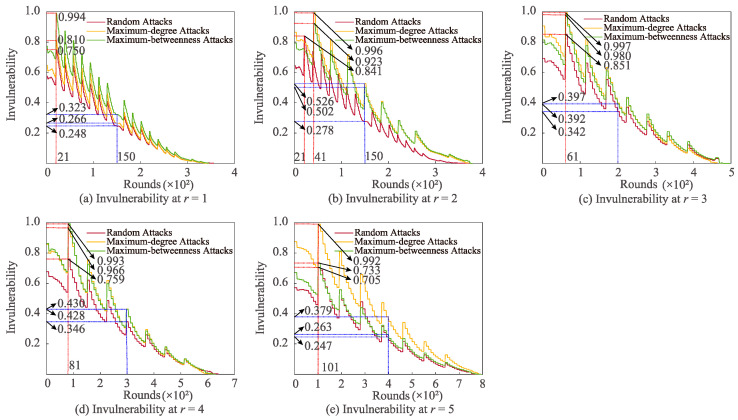
Impacts of altering *r* on invulnerability for various attacks.

**Figure 5 sensors-22-07936-f005:**
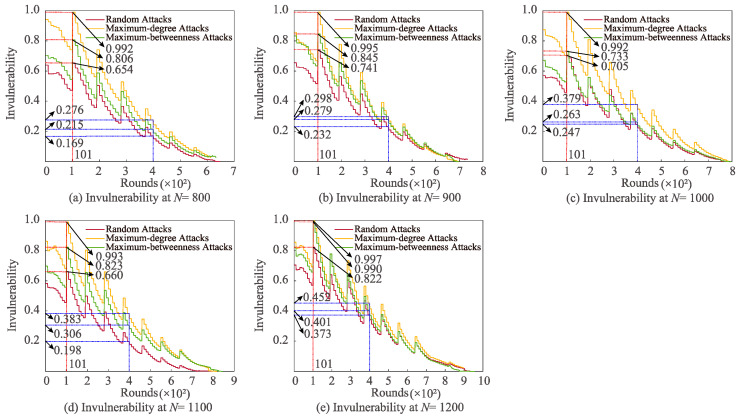
Influences of varying *N* on invulnerability for different attacks.

**Figure 6 sensors-22-07936-f006:**
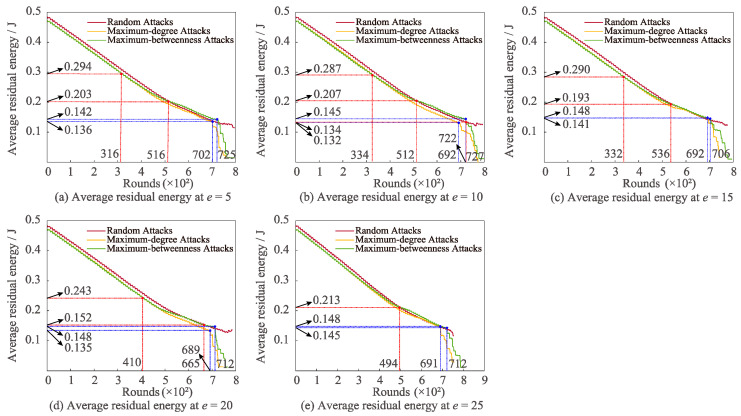
Relationships between differing *e* and average residual energy under distinct attacks.

**Figure 7 sensors-22-07936-f007:**
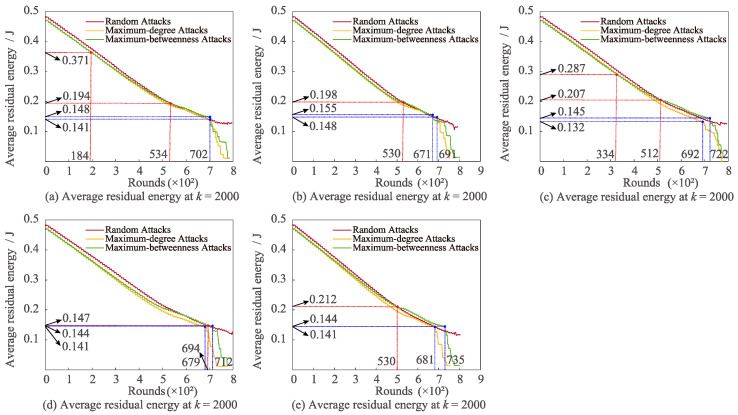
Effects of changing *k* on average residual energy for varied attacks.

**Figure 8 sensors-22-07936-f008:**
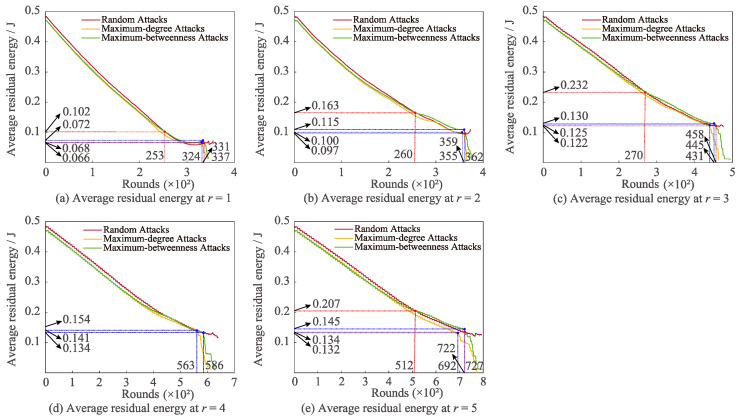
Impacts of altering *r* on average residual energy for various attacks.

**Figure 9 sensors-22-07936-f009:**
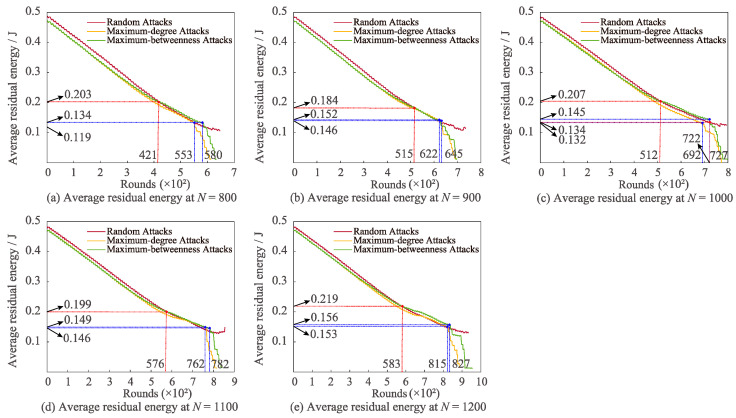
Influences of varying *N* on average residual energy for different attacks.

**Figure 10 sensors-22-07936-f010:**
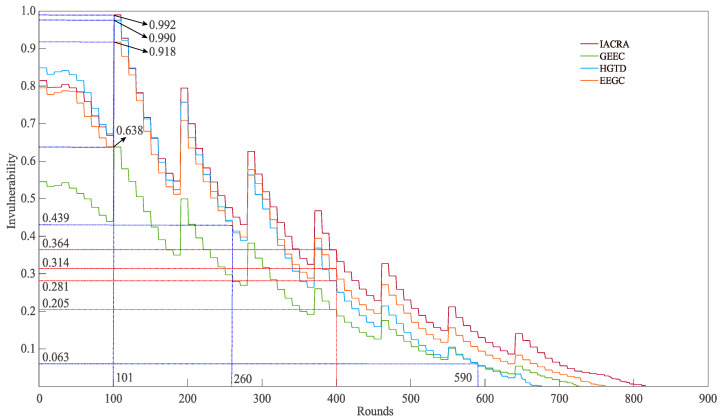
Comparisons of invulnerability of IACRA, GEEC, HGTD, and EEGC.

**Figure 11 sensors-22-07936-f011:**
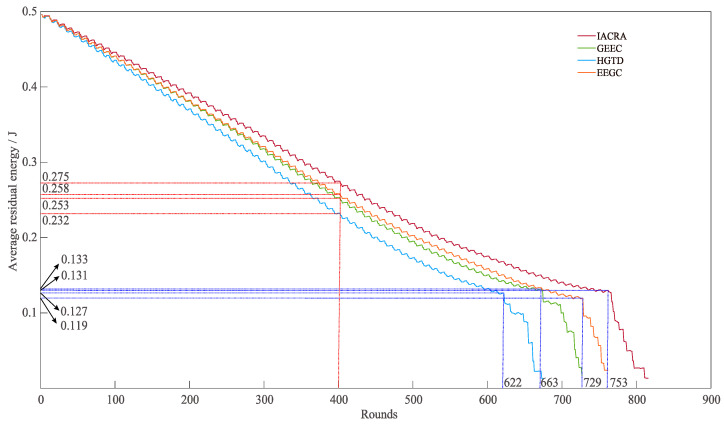
Comparisons of average residual energy of IACRA, GEEC, HGTD, and EEGC.

**Table 1 sensors-22-07936-t001:** Conventional routing protocols based on game theory in terms of technical strengths and weaknesses.

Protocol	Advantage	Limitation
GEEC	Reduced energy dissipation, location awareness	Generation of clusters with a single node
EEGC	Energy-related payoff definitions	Residual energy obviated from threshold function calculation, unoptimized transmission path after clustering
HGTD	Satisfactory trade-off between minimizing energy consumption and providing the required services	Residual energy excluded from the payoff definition

**Table 2 sensors-22-07936-t002:** Profit matrix.

Strategy	*D*	*ND*
*D*	$V\left(i\right)-C_{CH}\left(i\right)$,$V\left(i\right)-C_{CH}\left(i\right)$	$V\left(i\right)-C_{CH}\left(i\right)$,$V\left(i\right)-C_{CM}\left(i\right)$
*ND*	$V\left(i\right)-C_{CM}\left(i\right)$,$V\left(i\right)-C_{CH}\left(i\right)$	0,0

**Table 3 sensors-22-07936-t003:** Initial setting of simulation parameters.

Parameter	Value
*E*	0.5 J
*Eelec*	50 nJ/bits
*εfs*	10 pJ·bit^−1^·m^−2^
*εmp*	0.0013 pJ·bit^−1^·m^−4^
*Eaggr*	5 nJ/(bit/message)
Coordinates of the BS	(0,0)
*k*	4000
*N*	1000
*e%*	10%
*r*	5

## Data Availability

The data used to support the findings of this study are available from the corresponding author upon request.

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
