# Peer review of "IACRA: Lifetime Optimization by Invulnerability-Aware Clustering Routing Algorithm Using Game-Theoretic Approach for Wsns"

_sensors, 2022, doi:10.3390/s22207936_

Round 1
Reviewer 1 Report
1. Game theory has been widely used in network routing research. It is suggested to emphasize the innovation and advantages of this research in the Abstract.
2. In Section II, the authors should add the description of other models in the comparative experiment, and explain their characteristics, advantages and disadvantages. It is recommended to use a table to summarize several models in related work, which is more intuitive.
3. It is advised to modify Figure 1, reduce the size of the picture and enlarge the font to fill the blank space on page 10.
4. In Section 5.3, what is the reason for choosing the attack type? Please explain it. In addition to these two basic attacks in the network, is this algorithm applicable to other targeted attacks?
5. It is suggested that the author should add more information about the limitations of this study, which should be relevant to future work. For example, what will happen if adversaries are introduced into the game model, and how to solve the equilibrium.
6. The author's future work should be presented in the last section.
Reviewer 2 Report
The main content of research presented in the paper is a novel invulnerability-aware clustering routing algorithm using game-theoretic method proposed to solve the predicament.
The topic is not unique, but it is worthy of researching.
The main proposal is an invulnerability-aware clustering routing algorithm based on game-theoretic method to improve invulnerability and balance network energy consumption in wireless sensor networks.
The conclusions deduced based on the research methods are that the proposed protocol can be utilized to increase the capability of wireless sensor networks against deterioration of quality of service and energy constraints.
The conclusions are tenable. However, in the article it is not clear enough what progress has been made compared with the current research results.
The abstract is informative. It reflects the body of the paper.
The introduction provides sufficient background information for readers in the immediate field to understand the problem.
Overall, the text is well organized and the logic is clear. However, the text of the article would benefit greatly if it were proofread by a native English speaker. The related concepts are introduced clearly. The readability can be improved if authors follow the advice.
The novelty of the proposals is not sufficiently clear in the article. The differences between the functioning of the proposals and the traditional approaches are also not sufficiently clear.
The derivation of formulas and equations is rigorous enough.
The theoretical analysis is sufficient for the purposes of the article.
All figures and tables are clear enough to summarize the results for presentation to the readers. All figures and tables are well referred to in the text.
The reference section is informative. Overall, references are accurate. However, I advise authors to review the References section in order to make its formatting more complete, more homogeneous and in accordance with the journal's rules.
Round 2
Reviewer 1 Report
The author has revised the manuscript according to my requirements.
Reviewer 2 Report
The authors did a good job of reviewing and correcting the article. Now the article is ready for publication.